# *Phytophthora* Introductions in Restoration Areas: Responding to Protect California Native Flora from Human-Assisted Pathogen Spread

**Susan J. Frankel** [1,*] , **Christa Conforti** [2] , **Janell Hillman** [3] , **Mia Ingolia** [4] , **Alisa Shor** [5] , **Diana Benner** [6] , **Janice M. Alexander** [7] , **Elizabeth Bernhardt** [8] **and Tedmund J. Swiecki** [8]

1    USDA Forest Service, Pacific Southwest Research Station, 800 Buchanan Street, Albany, CA 94710, USA
2    Presidio Trust, 103 Montgomery St, San Francisco, CA 94129, USA; cconforti@presidiotrust.gov
3    Santa Clara Valley Water District, 5750 Almaden Expy, San Jose, CA 95118, USA; JHillman@valleywater.org
4    San Francisco Public Utilities Commission, 525 Golden Gate Ave., San Francisco, CA 94102, USA; MIngolia@sfwater.org
5    Golden Gate National Parks Conservancy, 201 Fort Mason, San Francisco, CA 94123, USA; AShor@ParksConservancy.org
6    The Watershed Nursery, 601A Canal Blvd, Richmond, CA 94804, USA; diana@thewatershednursery.com
7    University of California Cooperative Extension, Marin County, 1682 Novato Blvd, Suite 150B, Novato, CA 94947, USA; jalexander@ucanr.edu
8    Phytosphere Research, 1027 Davis Street, Vacaville, CA 95687, USA; phytoresearch@gmail.com (E.B.); phytosphere@phytosphere.com (T.J.S.)
*    Correspondence: susan.frankel@usda.gov

**Abstract:** Over the past several years, plantings of California native plant nursery stock in restoration areas have become recognized as a pathway for invasive species introductions, in particular *Phytophthora* pathogens, including first in the U.S. detections (*Phytophthora tentaculata*, *Phytophthora quercina*), new taxa, new hybrid species, and dozens of other soilborne species. Restoration plantings may be conducted in high-value and limited habitats to sustain or re-establish rare plant populations. Once established, *Phytophthora* pathogens infest the site and are very difficult to eradicate or manage—they degrade the natural resources the plantings were intended to enhance. To respond to unintended *Phytophthora* introductions, vegetation ecologists took a variety of measures to prevent pathogen introduction and spread, including treating infested areas by solarization, suspending plantings, switching to direct seeding, applying stringent phytosanitation requirements on contracted nursery stock, and building their own nursery for clean plant production. These individual or collective actions, loosely coordinated by the Phytophthoras in Native Habitats Work Group ensued as demands intensified for protection from the inadvertent purchase of infected plants from commercial native plant nurseries. Regulation and management of the dozens of *Phytophthora* species and scores of plant hosts present a challenge to the state, county, and federal agriculture officials and to the ornamental and restoration nursery industries. To rebuild confidence in the health of restoration nursery stock and prevent further *Phytophthora* introductions, a voluntary, statewide accreditation pilot project is underway which, upon completion of validation, is planned for statewide implementation.

**Keywords:** invasive plant pathogens; best management practices for phytosanitation; restoration nurseries

## 1. Introduction

California is home to exceptional botanical diversity. The California Floristic Province is recognized as a global biodiversity hot spot—an area with more than 1500 endemic plants and less than 30% of its original natural vegetation remaining [1,2].

Ecological restoration has the potential to increase biodiversity and deliver important ecosystem services. Defined as "the process of assisting the recovery of an ecosystem that has been degraded, damaged, or destroyed" [3], ecological restoration is most commonly conducted in response to human-caused habitat loss or degradation associated with infrastructure construction (e.g., roads, dams, utility corridors), invasive species introductions, and climate change. Restoration commonly involves intensive and costly ecological alterations through the addition and removal of species or barriers to connectivity [4]. Projects may be conducted on lands of high ecological value: habitats that are currently of limited extent due to an unusual combination of environmental conditions [5] or loss due to development.

In California, native plant nursery stock is the most common source for plant materials used in restoration projects [6]. The seed is field-collected and used to propagate plants in "restoration nurseries" that grow plants under contract for land managers. Plants for landscape use may be acquired from "native plant nurseries" that grow an array of indigenous plants on speculation for purchase by contractors, homeowners, or others attracted to pollinator promoting, environmentally-friendly gardens. Native plant nursery stock may be brokered or retailed by third parties and then used for restoration purposes. Additionally, native plant nurseries may procure plants from horticultural nurseries to fill out orders.

We highlight three case studies of restoration activities carried out within the past decade that have inadvertently introduced *Phytophthora* pathogens into high-value habitats. The case studies describe projects conducted in the San Francisco Bay Area by public agencies. Two of the case studies involve plantings required as mitigation. Both U.S. federal and state laws (e.g., the Federal and State Endangered Species Acts, Section 404 of the Clean Water Act, California Environmental Quality Act, etc.) require mitigation for construction projects and other activities that degrade or destroy sensitive habitats. Large construction projects are tightly regulated and subject to permitting by the California State Water Resources Control Board, California Department of Fish and Wildlife, U.S. Fish and Wildlife Service, U.S. Army Corps of Engineers, and others, dependent upon the location and project type. Progress toward habitat establishment is monitored and permitees may be fined if they do not meet agreed-upon success criteria by a specified time.

*Phytophthora* is a genus containing destructive plant pathogenic fungal-like organisms, with over 140 species [7]. *Phytophthora* introductions in restoration sites and California native plant nurseries were first recognized in 2012 to 2014 when unexpectedly, *Phytophthora tentaculata* Kröber and Marwitz was detected in Bay Area restoration areas on outplanted nursery stock of sticky monkeyflower (*Diplacus aurantiacus* (Curtis) Jeps.), toyon (*Heteromeles arbutifolia* (Lindl.) M. Roem.), coffeeberry (*Frangula californica* (Eschsch.) A. Gray), and sage (*Salvia* spp. L.) [8,9]. These plants are common native perennials that are frequently used in restoration plantings. The detections were first records for the U.S. Currently, *P. tentaculata* is included as a quarantine organism on the "U.S. Regulated Plant Pest List".

In another first U.S. detection, *P. quercina* T. Jung was found in a restoration area on a planted valley oak (*Quercus lobata* Née) in San Jose (Santa Clara County) [10]. Further sampling in restoration planting sites and native plant nurseries resulted in the detection of numerous new *Phytophthora* taxa and dozens of known *Phytophthora* species [10–12]. *Phytophthora* species are a common problem in horticultural nurseries [13–16], but *Phytophthora* species in California native plant or restoration nurseries were largely overlooked before the restoration site detection of *P. tentaculata* in 2014 [17]. Many introductions in restoration plantings occurred in the same region damaged by *P. ramorum* Werres, de Cock and Man in't Veld. Nearly 50 million native trees in California and Oregon have been

killed by *P. ramorum* since its introduction on nursery stock, estimated to have occurred in the 1980s to early 1990s [18].

Invasive species introductions are environmental threats that could occur mostly anywhere. We present our experiences of restoration site *Phytophthora* introductions and subsequent remediation efforts to draw attention to this pathway for pathogen movement and the benefits of prevention, primarily achieved by the use of clean planting stock. The case studies presented here illustrate the difficulties and high financial and ecological costs associated with inadvertent introductions highlighting the benefits of proactive steps to avert pathogen introductions.

## 2. Case Studies of Restoration Projects, *Phytophthora* Detections, and Response

### 2.1. Case Study 1. Restoration Area Phytophthora Management Conducted by the San Francisco Public Utilities Commission

The San Francisco Public Utilities Commission (SFPUC) provides power to San Francisco public services, wastewater treatment to San Francisco residents, and water for 2.6 million customers in four Bay Area counties via the Hetch Hetchy Regional Water System. In 2009, the SFPUC initiated a habitat restoration program to mitigate impacts associated with the Water System Improvement Program (WSIP), a voter-approved, $4.8 billion program that includes 87 capital improvement projects. The SFPUC Bioregional Habitat Restoration (BHR) program addresses the impacts of several WSIP construction projects by implementing a suite of habitat improvement projects that includes the development of compensation sites to preserve, enhance, or restore over 800 ha of watershed land. BHR projects cover 2.5 ha of ponds, 10 ha of seasonal wetlands, 6.4 km of stream systems, 40 ha of woodlands, and close to 730 ha of grasslands on lands owned by the SFPUC.

In the mid-2000s, SFPUC biologists observed coast live oaks (*Quercus agrifolia* Née) and tanoaks (*Notholithocarpus densiflorus* (Hook. and Arn.) P.S. Manos, C.H. Cannon, and S.H. Oh) dying in large numbers from the pathogen *P. ramorum* (causal agent of sudden oak death) on SFPUC managed watershed lands near Crystal Springs Reservoir (San Mateo County) [19–21]. In response, the SFPUC funded several forest health research projects in collaboration with the U.S. Forest Service. The projects uncovered numerous root-rotting *Phytophthora* species on watershed lands including *P. cinnamomi* Rands, *P. cambivora* (Petri) Buisman, and *P. cactorum* (Lebert and Cohn) J. Schröt that were associated with a long list of common native California plant species (e.g., Pacific madrone (*Arbutus menziesii* Pursh), California laurel (*Umbellularia californica* (Hook. and Arn.) Nutt.), and coffeeberry (*Frangula californica*)) [22].

To protect watershed lands from further infestations, the SFPUC imposed strict requirements on contractors and commercial growers that provided plants for the BHR restoration projects. The new specifications stipulated that the contractors were to provide "pest and pathogen-free" materials. This included cleaning heavy equipment of any soil or plant debris prior to delivery and heat treatment of large rootwads prior to placement in stream restoration. All of the contracted nurseries were given best practices guidance via the Oregon Association of Nurseries, *Safe Procurement and Production Manual* [23], and consultants were hired to inspect the nurseries during plant production. The specifications were comprehensive, but their unfamiliarity made it difficult for most of the contractors and nurseries to wholly implement.

Despite the safeguards, in 2014, biologists observed sticky monkeyflower (*Diplacus aurantiacus*), toyon (*Heteromeles arbutifolia*), and other nursery stock dying in restoration areas shortly after outplanting. Sampling confirmed the presence of numerous *Phytophthora* species including *P. tentaculata*, which had only recently been detected in the U.S. [9]. In response, the SFPUC halted the use of container plants and the importation of organic materials (e.g., mulch or compost) on all projects and began an extensive watershed-wide sampling program to characterize the extent of the infestations. Planted nursery stock was sampled by digging up the root ball and collecting about 1-L volume of roots and associated nursery container mix with small amounts of intermingled site soil. Most samples were collected from nursery stock showing possible Phytophthora root rot symptoms (dead, wilted, or stunted), but asymptomatic plants and empty planting basins were also sampled. In addition

to nursery stock sampling, samples were collected under dead and declining native vegetation at restoration sites and elsewhere on managed properties. Water samples were collected from runoff following winter rainstorms and from creeks and ponds. Most samples were baited for *Phytophthora* using green pears but some direct isolations and PCR probes of roots were also used for detection. Cultures were identified to species by the California Department of Agriculture Plant Pest Diagnostic lab or by plant pathologists at the Department of Plant Pathology, University of California, Davis using ITS sequences, sometimes in conjunction with COX2 sequences. Over 800 samples were collected between 2014 and 2019. Over 30 *Phytophthora* taxa on over 40 plant species were identified (Table 1).

**Table 1.** *Phytophthora* detections from samples collected on the San Francisco Public Utilities Commission (SFPUC) watershed lands in Alameda, San Mateo, and Santa Clara Counties between 2014 and 2019.

| *Phytophthora* Detected | Associated Hosts or Medium | # Positives |
|---|---|---|
| *Phytophthora amnicola* T.I. Burgess & T. Jung | Water | 1 |
| *Phytophthora bilorbang*/ P. taxon oaksoil Aghighi, G.E. Hardy, J.K. Scott & T.I. Burgess | Water | 2 |
| | *Quercus agrifolia* | |
| *Phytophthora borealis* E.M. Hansen, W. Sutton & Reeser | Water | 1 |
| *Phytophthora borealis* species complex | *Umbellularia californica* | 1 |
| *Phytophthora cactorum* (Lebert & Cohn) J. Schröt | *Acer macrophyllum* | 74 |
| | *Arbutus menziesii* | |
| | *Frangula californica* | |
| | *Heteromeles arbutifolia* | |
| | *Platanus racemosa* | |
| | Poaceae, species unidentified | |
| | *Quercus agrifolia* | |
| | *Quercus lobata* | |
| | Water | |
| | *Umbellularia californica* | |
| | Planting basin * | |
| *Phytophthora xcambivora* (Petri) Buisman | Arbutus menziesii | 67 |
| | *Baccharis pilularis* | |
| | *Ceanothus thyrsiflorus* | |
| | *Diplacus aurantiacus* | |
| | *Eriogonum nudum* | |
| | *Hesperocyparis macrocarpa* | |
| | *Heteromeles arbutifolia* | |
| | *Quercus agrifolia* | |
| | *Rubus ursinus* | |
| | *Toxicodendron diversilobum* | |
| | Water | |
| | *Umbellularia californica* | |
| | Planting basin * | |

**Table 1.** *Cont.*

| *Phytophthora* Detected | Associated Hosts or Medium | # Positives |
|---|---|---|
| *Phytophthora chlamydospora* Brasier & E.M. Hansen | *Artemisia douglasiana* | 25 |
| | *Diplacus aurantiacus* | |
| | *Frangula californica* | |
| | *Populus fremontii* | |
| | *Quercus agrifolia* | |
| | Water | |
| *Phytophthora cinnamomi* Rands | *Arbutus menziesii* | 32 |
| | *Quercus agrifolia* | |
| | *Umbellularia californica* | |
| *Phytophthora* aff. *citricola* | *Acer macrophyllum* | 1 |
| *Phytophthora citricola* species complex Sawada | *Acer macrophyllum* | 2 |
| | Poaceae, species unidentified | |
| *Phytophthora crassamura* B. Scanu, A. Deidda & T. Jung | *Platanus racemosa* | 4 |
| | Planting basin * | |
| *Phytophthora cryptogea* Pethybr. & Laff. | *Achillea millefolium* | 19 |
| | Annual forbs, unidentified | |
| | *Baccharis pilularis* | |
| | *Diplacus aurantiacus* | |
| | *Iris douglasii* | |
| | Poaceae, species unidentified | |
| | *Pseudotsuga menziesii* | |
| | *Solanum* sp. | |
| | *Toxicodendron* diversilobum | |
| | *Umbellularia californica* | |
| | Water | |
| *Phytophthora cryptogea* species complex | *Juncus* sp. | 10 |
| | *Quercus agrifolia* | |
| | *Diplacus aurantiacus* | |
| | Water | |
| *Phytophthora erythroseptica* Pethybr. | Water | 1 |
| *Phytophthora europaea* species complex E.M. Hansen & T. Jung | *Quercus agrifolia* | 2 |
| | *Rubus ursinus* | |
| *Phytophthora gonapodyides* (H.E. Petersen) Buisman | *Arbutus menziesii* | 54 |
| | *Artemisia douglasiana* | |
| | *Diplacus aurantiacus* | |
| | *Hordeum brachyantherum* | |
| | *Juncus* sp. | |
| | *Platanus racemosa* | |
| | *Quercus agrifolia* | |
| | *Salix* sp. | |
| | Water | |

**Table 1.** *Cont.*

| *Phytophthora* Detected | Associated Hosts or Medium | # Positives |
|---|---|---|
| *Phytophthora* taxon raspberry | Water | 2 |
| *Phytophthora inundata* Brasier, Sánch. Hern. & S.A. Kirk | *Euthamia occidentalis* | 7 |
| | *Juncus effusus* | |
| | *Juncus* sp. | |
| | Water | |
| *Phytophthora* sp. kelmania | *Diplacus aurantiacus* | 4 |
| | *Quercus agrifolia* | |
| *Phytophthora* aff. lacustris | Water | 4 |
| *Phytophthora lacustris* Brasier, Cacciola, Nechw., T. Jung & Bakonyi | *Platanus racemosa* | 11 |
| | Water | |
| *Phytophthora megasperma* species complex Drechsler | *Acer macrophyllum* | 37 |
| | *Arbutus menziesii* | |
| | *Artemisia douglasiana* | |
| | *Baccharis glutinosa* | |
| | *Conium maculatum* | |
| | *Diplacus aurantiacus* | |
| | *Euthamia occidentalis* | |
| | *Juncus balticus* | |
| | *Juncus patens* | |
| | *Platanus racemosa* | |
| | Poaceae, unidentified species | |
| | *Umbellularia californica* | |
| | Water | |
| *Phytophthora plurivora* T. Jung & T.I. Burgess | *Carex barbarae* | 1 |
| *Phytophthora quercetorum* Y. Balci & S. Balci | *Quercus agrifolia* | 4 |
| *Phytophthora riparia* Reeser, Sutton & E.M. Hansen | *Quercus agrifolia* | 4 |
| | Water | |
| *Phytophthora riparia* × *lacustris* | Water | 8 |
| *Phytophthora* spp. | *Heteromeles arbutifolia* | 13 |
| | *Mimulus aurantiacus* | |
| | *Quercus douglasii* | |
| | *Scrophularia californica* | |
| | Water | |
| *Phytophthora* taxon "cactorum-like" haplotype 1 | *Eucalyptus globulus, Heteromeles arbutifolia, Toxicodendron diversilobum* | 1 |
| *Phytophthora* taxon agrifolia | *Quercus agrifolia* | 1 |

**Table 1.** *Cont.*

| *Phytophthora* Detected | Associated Hosts or Medium | # Positives |
|---|---|---|
| *Phytophthora tentaculata* Kröber & Marwitz | *Artemisia douglasiana* | 14 |
| | *Frangula californica* | |
| | *Heteromeles arbutifolia* | |
| | *Diplacus aurantiacus* | |
| *Phytophthora thermophile* T. Jung, M.J.C. Stukely & T.I. Burgess | *Diplacus aurantiacus* | 1 |
| | Total: | 408 |

\* Planting basins–Plant had died or was removed, sampled soil and roots.

The restoration introductions ushered in a new era of land management for the SFPUC that stressed *Phytophthora* management and prevention. For revegetation, BHR switched to direct seeding, a lower disease risk activity in comparison to planting container stock. However, this created several new challenges since many wetland and small-seeded species proved to be very difficult to successfully establish via direct seeding.

Thousands of potentially *Phytophthora*-infected restoration plants were removed. Solarization and soil steaming were applied to contaminated restoration sites to control or eliminate the infestations with only partial success (Figure 1). Stricter biosecurity measures were implemented to prevent additional introductions and minimize pathogen spread. The new measures included heat treatment of organic materials before importation and a decontamination Standard Operating Procedure for tools, personal protective equipment, and vehicles that applies to anyone entering the watersheds.

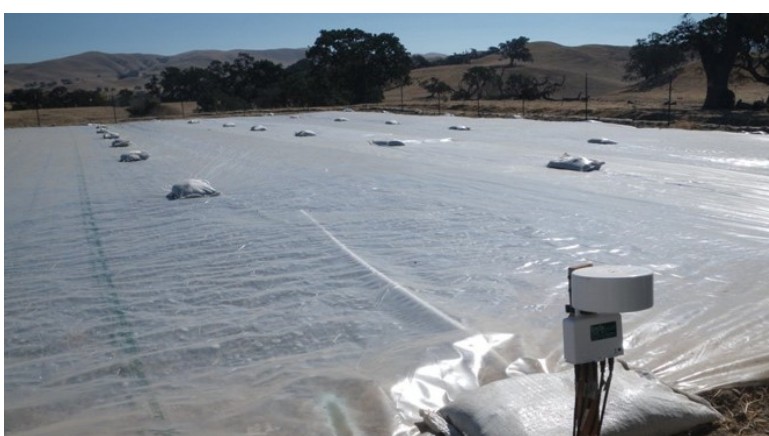

**Figure 1.** Solarization of a plant holding area near a large restoration planting where plants infected with *Phytophthora tentaculata* and other *Phytophthora* species were maintained for an extended period before planting. Soil temperature logger is at right. Credit: SFPUC.

In 2017, the need for landscape plantings at wildland-adjacent SFPUC facilities led the agency to construct a temporary nursery to propagate over 70,000 plants (Figure 2). The nursery was designed using the best management practices for restoration nurseries outlined by The Phytophthoras in Native Habitats Work Group [24] and incorporated all-metal growing surfaces; frequent shoe, surface, and tool sanitization; heat treatment of potting media; and a rigorous *Phytophthora* testing program to ensure nursery stock cleanliness. The nursery has tested *Phytophthora*-free since operations began in 2018, and SFPUC is considering expansion of the propagation efforts to help fulfill the needs of their restoration obligations.

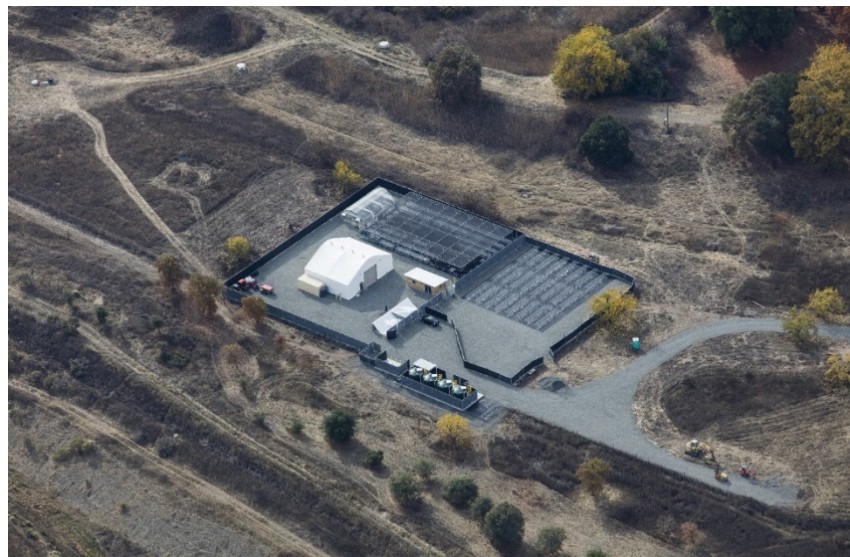

**Figure 2.** The SFPUC Sunol Native Plant Nursery in Alameda County. The design, layout, and operations of the facility incorporate strict phytosanitary measures to prevent *Phytophthora*. Credit: SFPUC.

### *2.2. Case Study 2. Phytophthora Awareness, Response, and Management for Endangered Species in the Presidio, Golden Gate National Recreation Area*

Within the Golden Gate National Recreation Area (GGNRA) is the Presidio, a 5666 ha urban National Park in San Francisco that is managed by the National Park Service and the Presidio Trust. The Presidio is highly developed with recreation areas, trails, a golf course, housing, hotels, office buildings, and museums. It is managed in three landscape zones, (1) Forest: areas where trees have been planted and maintained to create windbreaks and delineations; (2) Designed landscape: developed areas with ornamental plantings; and (3) Native habitat: areas maintained for native plant species. Formerly a military fort, the Presidio's forest, and designed landscape zones have been regularly planted with nursery-grown plants for over a century, and much of the native habitat zone has been restored using nursery-grown plants since the late 1990s.

The risk that *Phytophthora* could be introduced via nursery-grown plants to GGNRA sites with rare and endangered plants became apparent in 2014 when land managers learned of *Phytophthora* infestations in SFPUC restoration sites. In response to this concern, the GGNRA established a *Phytophthora* management team of land managers, scientists, restorationists, integrated pest management specialists, and nursery professionals. The initial focus was on the GGNRA native plant nurseries operated by the Golden Gate National Parks Conservancy. These nurseries provide approximately 150,000 container plants, annually, for habitat restoration projects throughout the GGNRA and nearly all the plant material for the Presidio native habitat zone.

When testing of GGNRA nurseries plant stock in late 2014 revealed the presence of several *Phytophthora* species, infected plant lots were discarded and all nursery growing spaces were deep-cleaned and sanitized. In early 2015, best management practices (BMPs) were integrated at each of the nurseries to reduce the risk of harboring *Phytophthora* and prevent further introductions into the field. The BMPs, which are strictly adhered to, include sterilization of reused plant containers, heat treatment of potting soil, drainage improvements in growing areas, sterilization of footwear, tools, and equipment, and additional routine sanitation measures. Quality control testing was initiated in 2015 to track the success of the nursery BMPs for *Phytophthora* control. *Phytophthora* detections were dramatically reduced within the first year of BMPs implementation. *Phytophthora* was at non-detectable levels at all nursery locations from 2016 through 2019, with a single detection in 2020. The infested lot was discarded and using trace-back information, the introduction pathway

was identified and eliminated. Monitoring remains an ongoing activity, including routine visual assessments, leachate baiting monitoring (http://phytosphere.com/BMPsnursery/test3_4bench.htm), and Agdia, ImmunoStrip® for *Phytophthora* and Pocket Diagnostic® *Phytophthora* rapid test. (Figure 3).

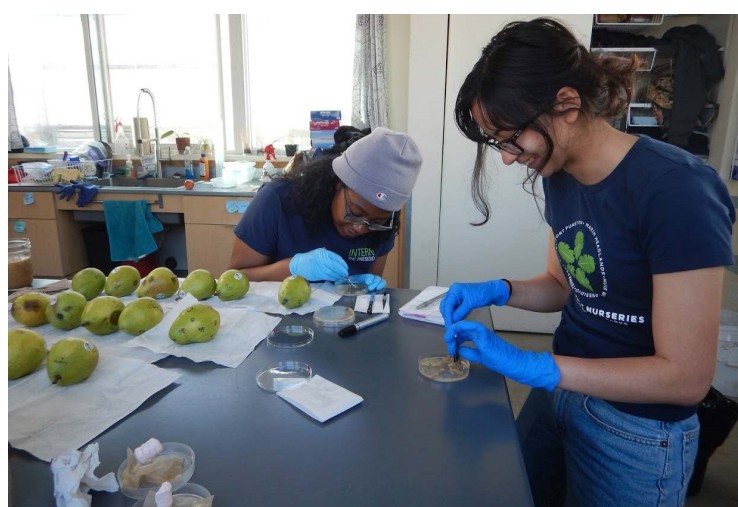

**Figure 3.** Interns isolating from pear baits to check for *Phytophthora* at the Presidio Nursery managed by the Golden Gate National Parks Conservancy (GGNPC). Credit: Alisa Shor, GGNPC.

Once GGNRA nursery BMPs were fully operational, the Presidio Trust turned its focus to determining the extent of *Phytophthora* in the Presidio landscapes, and to developing strategies for risk management. Special attention was given to habitat restoration sites containing rare and endangered plants that had previously been planted with nursery plants, including two endangered *Arctostaphylos* (manzanita) species, a genus known to be very susceptible to *Phytophthora* damage.

From 2015 to 2019, a baseline *Phytophthora* survey was conducted throughout the Presidio (Figure 4). Over four years, 1124 root and soil samples were tested, from 57 sites, targeting symptomatic, woody, or susceptible host species. In-house pear baiting was done and pears showing lesions were sent for species identification to the UC Berkeley Forest Pathology and Mycology Laboratory, or the California Department of Food and Agriculture (CDFA), Plant Disease Diagnostic Laboratory for morphological or molecular species identification, including isolation on selective media (pimaricin + ampicillin + rifampicin + pentachloronitrobenzene cornmeal agar, PARP), PCR, and occasionally ELISA tests. Seventeen percent of samples tested positive for *Phytophthora* and 22 *Phytophthora* species were recovered (Table 2). *P. citricola* complex, *P. cryptogea* Pethybr. and Laff., and *P. nicotianae* Breda de Haan were most common, accounting for 51% of all detections. Eighty-eight percent of sites had at least one *Phytophthora* species detection. However, detections did not occur evenly throughout or among sites. Wetter areas with natural or irrigation water had the highest detection levels, and drier areas with sandy soils had the lowest detection levels. Initially, areas around rare and endangered woody species had no detections.

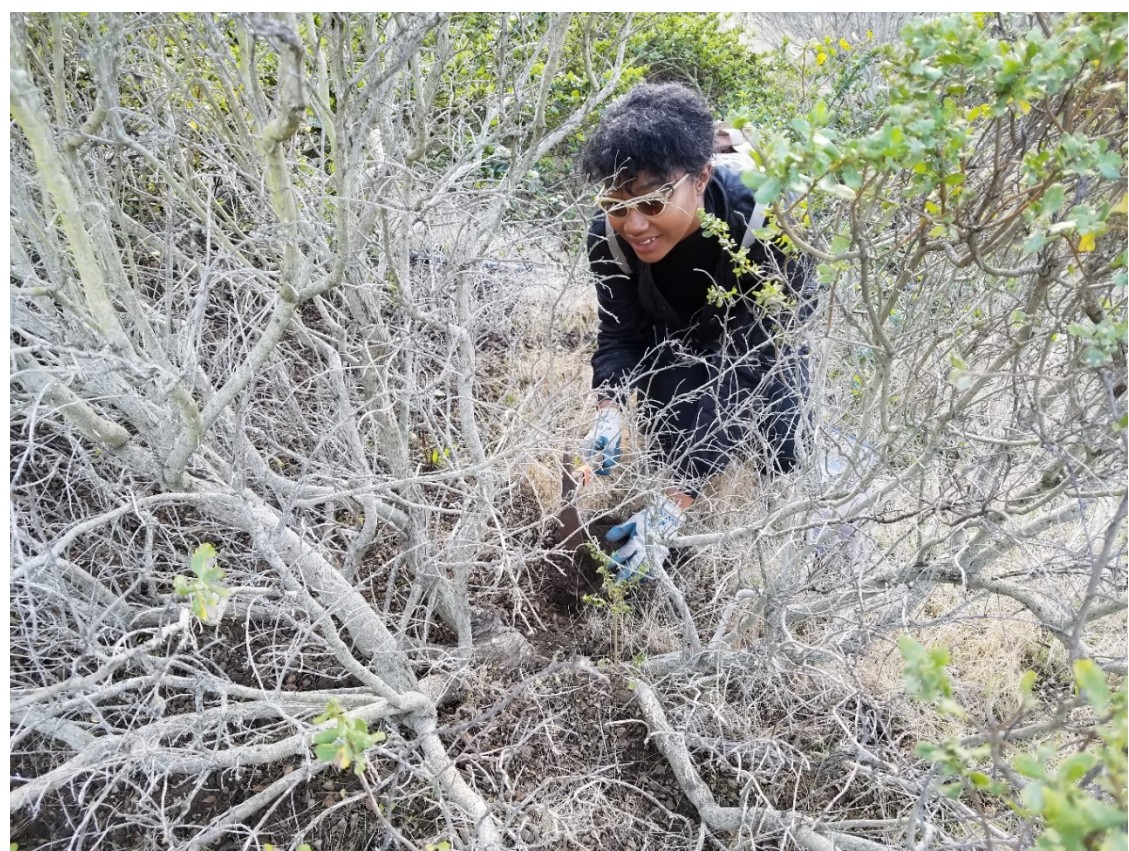

**Figure 4.** Sampling for *Phytophthora* at the Presidio. Credit: Courtesy of The Presidio Trust.

**Table 2.** Presidio of San Francisco *Phytophthora* detections.

| *Phytophthora* Species | Detections in Presidio Landscape (Total Number Samples Tested = 1124) | Detections on Incoming Nursery Plants (Total Number of Plant Lots Tested = 278) | Associated Hosts |
|---|---|---|---|
| *Phytophthora acerina* B. Ginetti, T. Jung, D.E.L. Cooke, S. Moricca | 1 | 1 | *Abelia grandiflora, Loropetalum* 'purple majesty' |
| *Phytophthora amnicola* T.I. Burgess and T. Jung | 1 | 0 | *Salix* sp. |
| *Phytophthora cactorum* (Lebert and Cohn) J. Schröt | 17 | 7 | *Buxus* sp., *Ceanothus thyrsiflorus, Cordyline australis, Fremontodendron californicum, Heteromeles arbutifolia, Juncus effusus, Juncus patens, Juniperus* sp., *Leptospermum laevigatum, Photinia fraseri, Pinus muricata, Pinus radiata, Prunus carolinia, Raphiolepis indica, Raphiolepis umbellata* 'minor', *Ribes sanguineum* |

**Table 2.** *Cont.*

| *Phytophthora* Species | Detections in Presidio Landscape (Total Number Samples Tested = 1124) | Detections on Incoming Nursery Plants (Total Number of Plant Lots Tested = 278) | Associated Hosts |
|---|---|---|---|
| *Phytophthora xcambivora* (Petri) Buisman | 12 | 3 | *Abelia grandiflora, Arbutus* 'Marina', *Cotoneaster* sp., *Heteromeles arbutifolia, Pinus contorta, Prunus caroliniana, Quercus agrifolia, Rumohra adiantiformis* |
| *Phytophthora chlamydospora* Brasier and E.M. Hansen | 3 | 2 | *Buxus japonica, Juncus patens, Leucodendron* sp. |
| *Phytophthora cinnamomi* Rands | 4 | 9 | *Agonis flexuosa* 'Jervis Bay Afterdark', *Arbutus* sp., *Azalea* sp., *Choisya* sp., *Hydrangea quercifolia, Lecuodendron* sp., *Leptospermum laevigatum, Osmanthus delavayi, Persea americana* |
| *Phytophthora citricola* complex (*P. citricola/P. multivora*) | 27 | 10 | *Abelia* 'Sherwoodii', *Agonis flexuosa, Alnus cordata, Baccharis pilularis, Ceanothus thyrsiflorus, Choisya ternata, Cistus salvifolius, Lepotospermum laveigatum, Prunus carolinia compacta, Quercus agrifolia, Rosmarinus officinalis, Tracholospermum jasminoides* |
| *Phytophthora crassamura* B. Scanu, A. Deidda and T. Jung | 6 | 0 | *Artemisia californica, Juncus patens, Lonicera involucrata, Rosa* sp., *Stuckenia pectinata* |
| *Phytophthora cryptogea* Pethybr. and Laff. | 49 | 14 | *Acacia melanoxylon, Acacia verticillata, Arbutus* sp., *Camelia* sp., *Ceanothus thrysiflorus, Choisya ternata, Cistus incanus, Correa* 'Ivory Bells', *Cupressus macrocarpa, Diplacus aurantiacus, Eriophyllum stachaedifolium, Eucalyptus globulus, Grevillea* 'Coastal Gem', *Grevillea rosmainifolia, Hardenbergia* sp., *Hedera helix* 'Hans', *Heteromeles arbutifolia, Hydrangea quercifolia, Juncus effsus, Juncus patens, Lantana sellowiana* 'Monma', *Leptospermum laevigatum, Myoperum laetum, Pinus radiata, Rosmarinus* 'Lockwood DeForest', *Rosmarinus officianalis, Rosmarinus officinalis prostratus, Rosmarinus officinalis* 'Upright', *Tristania elegant* |

**Table 2.** *Cont.*

| *Phytophthora* Species | Detections in Presidio Landscape (Total Number Samples Tested = 1124) | Detections on Incoming Nursery Plants (Total Number of Plant Lots Tested = 278) | Associated Hosts |
|---|---|---|---|
| *Phytophthora drechsleri* Tucker | 0 | 2 | *Grevellia* 'Penola', *Rosmarinus officinalis* |
| *Phytophthora gonapodyides* (H.E. Petersen) Buisman | 7 | 1 | *Abelia grandiflora, Arbutus* 'Marina', *Eucalyptus globulus, Frangula califormia, Juncus patens, Salix* sp., *Stuckenia pectinata* |
| *Phytophthora hedraiandra* De Cock and Man in 't Veld | 4 | 1 | *Abelia grandiflora, Artemisia californica, Juniperus horizontalis, Plumbago auriculata, Prunus* sp. |
| *Phytophthora inundata* Brasier, Sánch. Hern. and S.A. Kirk | 5 | 0 | *Carex densa, Heteromeles arbutifolia, Juncus lescurii, Juncus xiphioides, Salix* sp. |
| *Phytophthora* sp. kelmania | 3 | 1 | *Ceanothus thyrsiflorus, Choisya ternata, Juncus patens, Rosmarinus officinalis* |
| *Phytophthora lacustris* Brasier, Cacciola, Nechw., T. Jung and Bakonyi | 2 | 0 | *Juncus effusus, Salix* sp. |
| *Phytophthora megasperma* Drechsler | 4 | 1 | *Correa* 'Dusky Bells', *Diplacus aurantiacus, Eucalyptus citridora, Ligustrum* sp. |
| *Phytophthora nicotianae* Breda de Haan | 4 | 14 | *Abelia grandiflora, Agonis flexuosa* 'Jervis Bay Afterdark', *Anemone* sp., *Arbutus* 'Marina', *Cordyline australias, Correa* 'Dusky Bells', *Fuchsia thymifolia, Lucadendron salignum, Rosamarinus officinalis* |
| *Phytophthora niederhauserii* Z.G. Abad and J.A. Abad | 0 | 1 | *Phormium* 'rainbow chief' |
| *Phytophthora occultans* Man in 't Veld and K. Rosend. | 1 | 0 | *Buxus microphylla* |
| *Phytophthora palmivora* (E.J. Butler) E.J. Butler | 0 | 1 | *Pittosporum crassifolium* 'Nana' |
| *Phytophthora parvispora* Scanu and Denman | 1 | 1 | *Choisya ternata, Hebe buxifolia* |

**Table 2.** *Cont.*

| *Phytophthora* Species | Detections in Presidio Landscape (Total Number Samples Tested = 1124) | Detections on Incoming Nursery Plants (Total Number of Plant Lots Tested = 278) | Associated Hosts |
|---|---|---|---|
| *Phytophthora pseudocryptogea* Safaief., Mostowf., G.E. Hardy and T.I. Burgess | 34 | 5 | *Araucaria araucana, Baccharis pilularis, Correa* 'Dusky Bells', *Cotoneaster* sp., *Diplicus aurantiacus, Escallonia* sp., *Grevellia lanigera* 'Coastal Gem', *Sutera cordata, Eriophyllum stachaedifolium, Festuca* 'Elijah Blue', *Hebe buxifolia, Heteromeles arbutifolia, Juncus effusus, Juncus lescurii, Juncus patens, Juncus phaeocephalus, Juncus xiphioides, Leptospermum laevigatum, Loropetalum* 'purple majesty', *Pittosporum tenufolium, Rhododendron* sp., *Rosmarinus officinalis, Salix* sp., *Salvia clevelandii* |
| *Phytophthora ramorum* Werres, De Cock and Man in 't Veld | 1 | 0 | *Azalea* sp. |
| *Phytophthora rosacearum* E.M. Hansen | 1 | 0 | *Heteromeles arbutifolia* |
| *Phytophthora siskiyouensis* Reeser and E.M. Hansen | 1 | 0 | *Alnus cordata* |
| *Phytophthora tropicalis* Aragaki and J.Y. Uchida | 0 | 1 | *Correa* 'Dusky Bells' |
| Total | 188 | 75 | |

Presidio Trust land managers were confident that the introduction of *Phytophthora* into these sites on plants grown by GGNRA nurseries was no longer occurring. However, the Presidio Trust purchases ornamental and forest tree container plants from commercial nurseries for planting in the forest or designed landscape sites and, in many cases, these drain to habitat restoration sites. A risk management system for using commercial nursery plants in landscapes was implemented which involves buying plants only from nurseries with strict pathogen prevention practices. When that is not possible, the Presidio tests incoming plant lots before outplanting into sites that drain to habitat restoration sites. If tested plant lots contain *Phytophthora* species not already documented in the Presidio landscape, or rated high-risk by CDFA (i.e., rated A, B, or Q by California), or documented as highly destructive in California native plant communities, those plant lots are rejected. Over five years, eighteen *Phytophthora* species have been detected, including two not previously documented in North America, *P. parvispora* Scanu and Denman, and *P. acerina* B. Ginetti, T. Jung, D.E.L. Cooke, S. Moricca. The most commonly detected species were *P. citricola* complex, *P. cryptogea,* and *P. pseudocryptogea* Safaief., Mostowf., G.E. Hardy and T.I. Burgess, accounting for 59% of all detections. Overall, *Phytophthora* detection was common, found 75 times on about 27% of the plant lots (Table 2). Fifteen percent of the plant lots tested have been rejected.

Despite fairly extensive testing, the patterns of *Phytophthora* infestation remain problematic and difficult to track. *Phytophthora* precautionary measures continue to be implemented and improvements are needed. Particularly disconcerting, in the course of testing, *P. pseudocryptogea* was detected on an

endangered Raven's manzanita (*Arctostaphylos hookeri* G. Don ssp. *ravenii* P.V. Wells). The lone wild individual is showing significant dieback. When or how this rare manzanita became infected is not known but there is a history of planting in the area. Experimental phosphite treatments are being applied, along with the propagation of clean cuttings and planting clones. Best management practices are in place for field staff, including cleaning tools and boots when moving between sites, restrictions for off-road vehicles during the rainy season, as well as education for park users, outreach in volunteer programs, and trailside signage.

### 2.3. Case Study 3. Eradication of Phytophthora on an Endangered Species on Santa Clara Valley Water District Lands

As mitigation for impacts to the federally endangered Coyote ceanothus (*Ceanothus ferrisiae* McMinn) due to the planned Anderson Dam seismic retrofit, the Santa Clara Valley Water District (aka Valley Water, SCVWD) has been creating a new population of this chaparral shrub on a pristine mitigation property located on a ridge above the Anderson Reservoir near Morgan Hill, CA. Shortly after the first container plants, procured from a commercial restoration nursery, were installed at the site in 2014, decline and death of transplants became evident. Plants with and without top symptoms were sampled and baited as described above in case study 1. Sampled plants had root rot and were found to be almost universally infected with *P. cactorum* (22 of 24 samples). No *Phytophthora* species were recovered from nine samples collected from nearby native vegetation.

Starting in 2015, root sampling was conducted within an existing stand of Coyote ceanothus near Anderson Reservoir with areas of plant decline and mortality. Initial sampling detected several *Phytophthora* species near a 1993 restoration planting of nursery-grown *C. ferrisiae*. *Phytophthora* was detected in 65% (15 of 23) of the root samples collected from dead and declining plants in the 1993 planting basins and other plants in a 2.8 ha area centered around and extending downslope from the planting. Species detected included *P. cactorum*, *P.* x*cambivora*, *P. crassamura*, *P. syringae*, and *P.* sp. kelmania, all species which have been detected in nursery stock [12,25,26]. No *Phytophthora* was detected from 34 samples of *C. ferrisiae* and other plants located upslope or beyond the drainage area below the 1993 planting. The results support the hypothesis that the multispecies *Phytophthora* infestation in the area of the planting was likely to have originated with the planting of infected nursery stock in 1993 [27]. This information reinforced Valley Water's decision to try to eradicate *P. cactorum* from the mitigation site above the Anderson Reservoir.

Because the mitigation property includes a variety of high-value habitats such as grey pine (*Pinus sabiniana* Douglas ex Douglas) woodland, serpentine grassland, and mixed-sage chaparral, remediation of the several hundred infested plant basins at the site proved to be complicated. The site is remote and difficult to access during the winter months; all supplies must be brought up to the ridgeline via four-wheel-drive vehicles. However, the pristine nature of the site and documentation that no *Phytophthora* species were present in the native vegetation meant that full remediation to eradicate *P. cactorum* was imperative.

A conventional solarization method [28] was adapted for use in treating individual planting sites. After careful removal of the above-ground portion of infected plants, planting sites were covered with two square layers (1.2 m side length) of thermal anti-condensate greenhouse film, each of which was sealed to the ground. The film was left in place for more than a year and was cleaned periodically to optimize solar heating. Site sampling showed that the pathogen was not detectable within one year in treated sites in full sun. In areas with extensive shading from trees, the solarization of infested basins was unsuccessful. Temperature monitoring indicated that *P. cactorum* persisted where soil temperatures at 20 cm depth did not exceed 35 °C for at least 100 h.

To effectively remediate those basins, three solar ovens were constructed, brought to the site, and placed in adjacent sunny areas (Figure 5). Light meters were used to measure and record the light intensity of the solarized planting basins and to prioritize basins for additional remediation. Infested soil was carefully and laboriously excavated from each planting basin to a depth of 25–30 cm below the original planting site grade and placed in three, 5-gallon (19 L) metal buckets, positioned in a

solar oven, and wetted to field capacity. The buckets were covered with greenhouse film to retard evaporation and prevent condensation on the solar oven window. The soil temperature range in the buckets was tracked using data loggers. Once temperature–time treatment thresholds were exceeded (one day with 1 h ≥ 70 °C; two days with either ≥60 °C for 30 min or ≥50 °C for 90 min), the buckets of soil were carefully emptied back into their original planting basin. Each basin was then marked with a permanent survey marker. Solar oven remediation occurred over a three-month period in summer when solar radiation was at its peak, mostly with 3- to 4-day treatments. Treatment thresholds were exceeded by large margins for all treated basins.

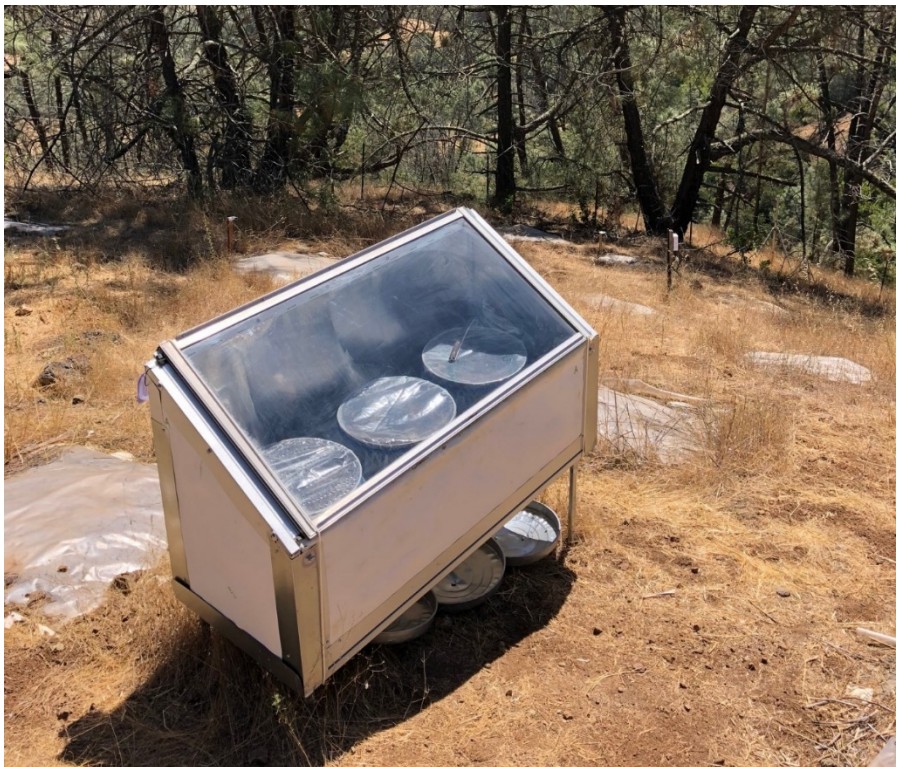

**Figure 5.** Solar oven used to heat treat soil from infested plant basins. Credit: Janell Hillman, SCVWD.

Complete site remediation (verified by soil sampling) was achieved by the fall of 2018, at a cost of approximately $900 per planting basin using solarization and approximately $1800 per planting basin for areas that needed the additional solar oven treatment.

Planting via direct seeding and container plant installation (following strict phytosanitary guidelines for growing, planting, and maintaining the nursery stock) has since resumed and to date, approximately 800 Coyote ceanothus plants have been installed at the mitigation site (Figure 6). The first documented flowering and seed production of the mitigation plants occurred in 2019, and the project is back on track to meet its ultimate success requirements and timeline to completion.

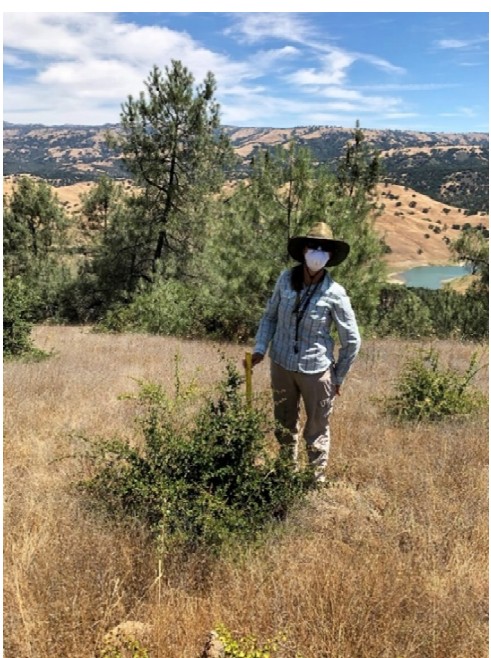

**Figure 6.** Healthy Coyote ceanothus growing after successful *Phytophthora* eradication at a mitigation site in Santa Clara County. Credit: Janell Hillman, Santa Clara Valley Water District.

The discovery of *Phytophthora* at the Coyote ceanothus mitigation site and near Anderson Reservoir led to additional studies of Valley Water's mitigation sites across Santa Clara County. Roots and associated soil of poorly performing outplanted container stock were collected and tested for *Phytophthora* species as described above in case study 1. In one study, *Phytophthora* species were isolated from 73 of 191 samples from 23 of 31 planting locations, yielding 38 species of *Phytophthora* [10]. Detections included *P. tentaculata* and *P. quercina* (not previously detected in the US), as well as undescribed *Phytophthora* taxa. In response, Valley Water initiated a Plant Pathogen Program, with a focus on improved sanitary practices for growing and planting mitigation plantings, as well as increased emphasis on equipment sanitation for construction and maintenance projects. The program has resulted in improved plant health and increased success of revegetation projects across the agency's watersheds.

## 3. Collective Action to Prevent *Phytophthora* Introduction and Spread in Restoration Areas

In 2014–2015, the *Phytophthora* detections described here along with others led some land managers to suspend plantings, cancel nursery stock orders, and attempt to clean up contaminated sites. To address the issues highlighted by these *Phytophthora* detections, the Phytophthoras in Native Habitats Work Group (www.calphytos.org) was formed in 2015. The Work Group is a cross-disciplinary, voluntary coalition dedicated to minimizing the spread of *Phytophthora* species on California native plants, with a focus on restoration site health. It brings together organizations to develop and share science-based technical assistance to improve and coordinate *Phytophthora* management, monitoring, research, education, and policy.

After learning about several *Phytophthora* introductions, the California restoration community was eager to prevent pathogen spread. Vegetation ecologists restoring sensitive habitats were still interested in using nursery stock if they could have confidence that the stock is *Phytophthora*-free. Many environmentally-oriented restoration nursery growers were highly motivated to produce the clean stock. To provide a framework for the production of nursery stock free of *Phytophthora* to the greatest practical degree, the Work Group and the California Native Plant Society developed comprehensive nursery best management practices (BMPs) for restoration nursery stock [24]. The BMPs are detailed, rigorous, and have the goal of excluding *Phytophthora* from nursery stock by "starting clean and staying clean". Following well-established principles of pathogen exclusion [29], the BMPs

require the use of clean inputs: sanitized or new containers, pathogen-free water supplies, heat-treated potting media, and plant propagules (e.g., seed) free of *Phytophthora.* "Staying clean" is accomplished by practices that prevent accidental contamination of plants and all associated inputs. Key components include maintaining all stock at least 60 cm above an improved ground surface on suitable benches, and monitoring and testing of stock by the nursery to detect possible *Phytophthora* contamination. The use of oomycete-suppressing pesticides that could mask infestations [30] is forbidden. To meet the BMP standards, growers need to dedicate a significant amount of effort and attention to phytosanitation throughout their operations; many nurseries have had to revise nursery layout, acquire equipment for heat-treating potting media and cleaning containers, adjust propagation practices, and provide additional training for workers.

In 2018, the BMPs were incorporated into "Accreditation to Improve Restoration and Native Plant Nursery Stock Cleanliness" (AIR), and a pilot project was initiated to determine whether an audit-based accreditation program could be used to document how nurseries were effectively implementing the BMPs. Such accreditation would allow clients to have confidence that plants produced by a nursery are not infected with *Phytophthora* species. About two dozen nurseries, located in northern and southern California, voluntarily enrolled. Each participating nursery completes an extensive online evaluation form that documents how they are implementing the BMPs, which is reviewed by AIR program evaluators. Evaluators then conduct a site visit to assess the nursery's infrastructure for risk pathways, discuss and clarify the nursery's reported practices, and conduct limited testing of nursery stock for the presence of *Phytophthora* using an irrigation leachate baiting method [31,32]. In this pilot phase, the emphasis has been on providing advice on ways that the nursery can address any BMP compliance issues within their site and operational constraints.

By following the AIR guidance, several nurseries have had no *Phytophthora* detections for two years or longer based on quality control testing by the nursery and, for some, extensive third-party predelivery testing of stock conducted for nursery clients. The AIR program continues to evolve and is slated to become a statewide voluntary certification program.

## 4. Discussion

The goal of restoration is to improve ecological function and health, but it is also a type of disturbance with many activities that pose risks for invasive species introductions, (e.g., planting, soil movement, use of heavy construction equipment, and frequent worker ingress and egress). Nursery stock outplanted into wildland settings presents an opportunity for pathogen infections acquired in nurseries to be moved into new locations along with the plants, which may inadvertently result in lasting environmental damage to surrounding natural communities and habitats [18]. In California, native plant nursery stock outplanted for restoration has served as a pathway of introduction for a diverse array of *Phytophthora* species into natural areas and critical habitats [12]. Starting with healthy, disease-free stock gives restoration projects the best chance of meeting established performance targets.

The incidence of *Phytophthora* disease in California native plant nurseries that do not adhere to strict clean production, BMPs can be substantial. A study by Sims et al. [25] reported disease incidences in five California restoration nurseries of 22% to 32%, which is similar to the 17% to 31% incidence reported for five commercial Oregon nurseries found by Osterbauer et al. [33]. Many *Phytophthora* species commonly occur in nurseries, and Parke et al. [15] detected 28 different *Phytophthora* taxa in four commercial nurseries.

Multiple *Phytophthora* species were detected at many of the restoration sites sampled in these case studies. Clearly, the risk of initiating a *Phytophthora* infestation at a restoration site is high when multiple *Phytophthora*-infected plants are planted and then irrigated for a year or more, as is typical in California restoration projects.

The risk associated with using *Phytophthora*-infected stock for restoration is controllable and avoidable. Following strict BMP guidelines has succeeded in reducing *Phytophthora* disease incidence

in participating nurseries to undetectable levels. A similar accreditation program, the Avocado Nursery Voluntary Accreditation Scheme (ANVAS), has been used successfully in Australia to produce avocado planting stock free of *P. cinnamomi* [34,35]. Both ANVAS and the AIR program are largely based on clean nursery production principles presented by Baker [29].

Outplanted nursery stock is not the only means by which restoration sites may become infested by *Phytophthora*. For example, *Phytophthora* may be present in materials used in plant installations, such as compost used as mulch [26]. Many restoration sites in California have previous land uses, such as cropland, orchards, Christmas tree farms, or other development, that may have residual *Phytophthora* infestations [11,27]. Urban areas, in particular, are known to harbor a number of *Phytophthora* species due to their history of horticultural plantings and diverse land uses [36–39]. *Phytophthora* can be spread from existing infestations to new locations by the incidental movement of infested soil and plant debris on vehicles and shoes, as the result of soil import, export, and grading, or via inoculum transported in surface waters. To address these issues in restoration and other land management activities, a framework for evaluating and mitigating risks associated with these pathways has been developed [40].

Local environmental conditions within the case study areas influence *Phytophthora* distribution and prevalence. At the Presidio, *Phytophthora* was more common in wet areas and low-lying areas prone to water inundation. This same pattern was also observed for *Phytophthora* infestations in Santa Clara habitats managed as habitat reserves [27]. However, *Phytophthora* infestations have been detected causing significant disease on native plants in dry upland sites in California [41].

Although an increasing number of *Phytophthora* species associated with the decline and mortality of woody plants have been recognized in California [42], we do not fully understand the extent of damage that many of the detected species cause on various host plants. California native plants have summer drought adaptations that may cause them to appear weak and off-color, which can make disease detection difficult. Sampling in these and related studies [10] have resulted in a large number of previously undescribed host–pathogen combinations. The baiting techniques [11] used for many of these investigations show associations between the host plant roots and *Phytophthora* species but do not prove that the organisms are causing disease. Koch's postulates have been completed for some of these pathogen–host associations (e.g., [8,9,12]), but more work is needed to fully understand the ecological impacts of these *Phytophthora*–host plant combinations. Even when pathogenicity is confirmed in controlled inoculations, disease expression in native stands is influenced by environmental and host factors.

Precautions are warranted to prevent *Phytophthora* introductions into restoration sites given the large number of highly damaging *Phytophthora* diseases that occur on a wide range of plants [43] and the ecological value of restoration investments. Inadvertent pathogen introductions into wildlands on nursery stock can start outbreaks that have a cascade of harmful effects resulting from plant death and decline. These include increases in fuel loads, soil erosion, increases in invasive plant cover, and degraded habitat for other species. There is no way to eradicate the pathogens once widely established [44], and negative effects may be long-term and generally irreversible as has been seen with introductions of *P. ramorum* [45] and *P. lateralis*, cause of Port-Orford-cedar root disease [46].

## 5. Conclusions

Although additional study is needed to track long-term impacts of *Phytophthora* diseases in restoration sites and other infested areas, preventing introductions provides the best means for avoiding *Phytophthora* disease impacts to native habitats and managed landscapes. Even if detected early and localized in planting basins, eradication of *Phytophthora* infections in outplanted nursery stock is technically difficult and often prohibitively expensive, especially if a large number of planting sites are involved. Using *Phytophthora*-free nursery stock prevents the introduction of these plant pathogens into restoration sites and results in healthier and more robust plants with better prospects for establishment, growth, and survival. To minimize risk, the use of *Phytophthora*-free planting material

coupled with best practices to avoid other sources of *Phytophthora* contamination (e.g., infested compost) is beneficial to protect wildlands. The case studies demonstrate that this nursery to wildland invasive species pathway of introduction can be disrupted—assisting land managers with the achievement of their restoration goals.

**Author Contributions:** Design and writing, S.J.F., T.J.S. and E.B.; source information, narratives, tables, figures provided by M.I., J.H., A.S. and C.C.; T.J.S., E.B., D.B. and J.M.A. designed or implemented many of the techniques and strategies described. All authors have read and agreed to the published version of the manuscript.

**Funding:** Funding was provided by the San Francisco Public Utilities Commission, Santa Clara Valley Water District (Valley Water), The Presidio Trust, and Golden Gate National Parks Conservancy.

**Acknowledgments:** The authors would like to thank Suzanne Rooney-Latham and Cheryl Blomquist, California Department of Food and Agriculture, Matteo Garbelotto and Laura Sims, University of California–Berkeley, and Tyler Bourret, UC Davis for species identifications and other diagnostic assistance. We also appreciate the support of the USDA Forest Service Pacific Southwest Research Station. Lastly, we recognize the University of California Cooperative Extension, David Lewis, Bonnie Nielsen, and Ana Medina for the critical program and administrative support they have provided.

**Conflicts of Interest:** The authors declare no conflict of interest.

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
