# Peer review of "Phytophthora Introductions in Restoration Areas: Responding to Protect California Native Flora from Human-Assisted Pathogen Spread"

_forests, doi:10.3390/f11121291_

Round 1

Reviewer 1 Report

Dear Authors,

The manuscript ID: forests-999276, entitled " Phytophthora Introductions in Restoration Areas:Responding to protect California Native Flora from Human-assisted Pathogen Spread" is original and meets the criteria required by journal Forests, the Special Issue "Role of Human Interventions in Spread of Soilborne Forest Pathogens and Methods for Mitigation".

The article provides very important and interesting information about the introductions of the harmful mainly soilborne forest Phytophthora pathogens including quarantine species (Ph. tenteculate) and dangerous for oak trees Ph. quercetorum, Ph. ramorum and Ph.cactorum during recovery and transfer of contaminated plant material to human-damaged or destroyed habitats associated with infrastructure construction in California. Currently, Phytophthora infestation of forest trees remains problematic worldwide and difficult to track. The article highlights three case studies of restoration activities that have inadvertently introduced Phytophthora pathogens into high-value habitats and discuss a variety of measures to prevent the introduction and spread of these pathogens.  The article also provides useful data, which can be used for the restoration and prevention of Phytophthora infection in damaged habitats, primarily through the use of clean planting material free of Phytophthora. In this article authors summarise the results of three projects carried out within the past decade in California and main outcomes in relation to Phytophthora risks identified and the human activities needed to mitigate these risks. All conclusions are supported by the results of these studies.

The article in principle can be accepted after minor revision. However, I have a few comments and questions, listed below.

p.3, Line 119. Phytophthora ramorum is not only aerial pathogen. I suggest correcting  "... pathogen P. ramorum..."

p.3, Line 142-143. Please clarify which samples were collected? (water, soil, plant roots, stems, leaves?) and describe the methods used to isolate and identify or confirm Phytophthora species.

p. 4, Line 144. Table 1 and Table 2. Please insert the author names tp Phytophthora species. Table 1. For some Phytophthora species with a wide range of hosts, you give only the total number of positive results. It will be more informative to enter data on positive results for each host species. These data show which hosts are more likely to be infected with a specific Phytophthora pathogens.

p.9, Line 206 "Monitiring remains an ongoing activity" . Please clarify what activity and what methods are used during monitoring and detecting Phytophthora pathogens. Line 210. Wtat does this mean? "...non-detectible levels".. Any positive results after testing 100 or more? soil or host plants samples?

p.10, Line 219. Please clarify  methods used for Phytophthora pathogens identification (cultural, morphological, molecular?).

p.15, Line 311. Please insert more data about detected Phytophthora species (possibly in a table like the one you provided for Case study 1 and Case study 2?)

p. 16, Line 316. Please clarify the numeration of manuscript sections. The section 3 is lost.

p. 18, Line 419-426. This paragraph will be better transfer to Discussion section, Line 417.

Reviewer 2 Report

This is a kind of review paper on the introduction of Phytophthora species in restoration sites in California mainly linked with nursery stocks. In general, the role of nurseries as a pathway for the introduction of Phytophthora in new areas is well studied and recognized. However, I disagree with the statement that Phytophthora biodiversity in a certain site is just the consequence of multiple introductions. Since many of the known Phytophthoras are cryptic species, It is extremely complicated to state if they are recently introduced or part of the long term site soil biodiversity. I commented on some sentences in the uploaded reviewed text specifically regarding the above considerations. Minor editing is also recorded as a comment. 

I strongly suggest the authors briefly discuss the above issue in the discussion section
